# N-Glycosylation as a Modulator of Protein Conformation and Assembly in Disease

**DOI:** 10.3390/biom14030282

**Published:** 2024-02-27

**Authors:** Chiranjeevi Pasala, Sahil Sharma, Tanaya Roychowdhury, Elisabetta Moroni, Giorgio Colombo, Gabriela Chiosis

**Affiliations:** 1Chemical Biology Program, Memorial Sloan Kettering Cancer Center, New York, NY 10065, USA; pasalac@mskcc.org (C.P.); sharmas5@mskcc.org (S.S.); roychot@mskcc.org (T.R.); 2The Institute of Chemical Sciences and Technologies (SCITEC), Italian National Research Council (CNR), 20131 Milano, Italy; elisabetta.moroni@scitec.cnr.it (E.M.); g.colombo@unipv.it (G.C.); 3Department of Chemistry, University of Pavia, 27100 Pavia, Italy; 4Department of Medicine, Memorial Sloan Kettering Cancer Center, New York, NY 10065, USA

**Keywords:** N-glycosylation, disease, conformational mutant, aberrant protein assembly, energy landscape, SARS-CoV-2 spike protein, glucose-regulated protein 94 (GRP94), prion protein, disease-associated protein conformation, protein dynamics, protein assembly mutation, gain-of-function conformational change

## Abstract

Glycosylation, a prevalent post-translational modification, plays a pivotal role in regulating intricate cellular processes by covalently attaching glycans to macromolecules. Dysregulated glycosylation is linked to a spectrum of diseases, encompassing cancer, neurodegenerative disorders, congenital disorders, infections, and inflammation. This review delves into the intricate interplay between glycosylation and protein conformation, with a specific focus on the profound impact of N-glycans on the selection of distinct protein conformations characterized by distinct interactomes—namely, protein assemblies—under normal and pathological conditions across various diseases. We begin by examining the spike protein of the SARS virus, illustrating how N-glycans regulate the infectivity of pathogenic agents. Subsequently, we utilize the prion protein and the chaperone glucose-regulated protein 94 as examples, exploring instances where N-glycosylation transforms physiological protein structures into disease-associated forms. Unraveling these connections provides valuable insights into potential therapeutic avenues and a deeper comprehension of the molecular intricacies that underlie disease conditions. This exploration of glycosylation’s influence on protein conformation effectively bridges the gap between the glycome and disease, offering a comprehensive perspective on the therapeutic implications of targeting conformational mutants and their pathologic assemblies in various diseases. The goal is to unravel the nuances of these post-translational modifications, shedding light on how they contribute to the intricate interplay between protein conformation, assembly, and disease.

## 1. Introduction

Glycosylation stands as a fundamental and pervasive post-translational modification crucial to various cellular processes. This complex process involves the covalent attachment of carbohydrates, collectively known as glycans, to macromolecules, shaping their structure and function. Orchestrated by a diverse set of enzymes—glycosyltransferases and glycosidases—encoded by a significant fraction of the human genome, glycosylation plays a central role in cellular regulation [1,2,3,4].

The importance of glycosylation is underscored by the abundance of proteins dedicated to shaping glycans. Over 750,000 glycosyltransferase sequences, distributed across all kingdoms, have been identified, and this number is steadily growing [5]. These enzymes, categorized into more than 110 glycosyltransferase families in the Carbohydrate-Active Enzymes (CAZy) database (http://www.cazy.org, accessed on 9 October 2023), exemplify the diversity and complexity of glycan modification. Glycosidases, comprising over 870,000 members in more than 170 CAZy families, contribute to the dynamic turnover of glycans [5].

There are a variety of building blocks for the generation of glycans [6,7,8]. In mammals, glycans are composed of a repertoire of 10 monosaccharides—D-galactose (Gal), D-glucuronic acid (GlcA), D-glucose (Glc), D-mannose (Man), D-ribose (Rib), D-xylose (Xyl), L-fucose (Fuc), N-acetyl-D-galactosamine (GalNAc), N-acetyl-D-glucosamine (GlcNAc), and N-acetylneuraminic acid (Neu5Ac). These monosaccharides form linear or branched structures through α- or β-glycosidic bonds, constituting the glycan repertoire observed in various cellular components [6,7,8].

Glycosylation extends its influence across multiple macromolecules, including proteins, lipids, and nucleic acids [9]. The complement of glycan structures produced by cells is referred to as the glycome (in analogy to the genome, transcriptome, proteome, lipidome, and metabolome). Glycans attached to lipid molecules form glycolipids. These are important components of cell membranes, and they are involved in cell recognition and signaling. Recently, it has been discovered that RNA molecules, particularly small, non-coding RNAs, can undergo glycosylation, uncovering new roles for glycosylation in RNA function [9]. Glycoproteins, however, formed by the attachment of glycans to proteins constitute a major part of the glycome [9].

Protein glycosylation is characterized by two major linkages: N-linked glycosylation, involving the amide nitrogen of asparagine residues, and O-linked glycosylation, linking carbohydrates to the hydroxyl oxygen of serine, threonine, or tyrosine residues. N-glycans are attached to asparagine residues within specific sequons (i.e., a consensus Asn-XSer/Thr where X is any amino acid except proline) [10]. In contrast, O-glycans are commonly linked to serine or threonine residues, including variations, such as O-linked GlcNAc and mucin-type O-glycans. Unlike N-glycans, O-GalNAc-linked glycans lack a specific glycoside amino acid sequon [10].

Glycosylation of proteins is a non-templated process regulated by enzymes, specifically glycosyltransferases, which initiate or elongate glycans, and oligosaccharyltransferases, which add entire carbohydrate chains [1,3,4]. In the context of biological processes, a template is often a molecule or a sequence of information that guides the assembly or formation of a particular structure (i.e., DNA to protein). In contrast, non-templated processes lack a strict template-based guidance system. In this sense, glycosylation exhibits a degree of flexibility in the generation of products, a process influenced by the cellular milieu and its response to intracellular and extracellular cues. In cells, the complex interplay between glycosyltransferases or oligosaccharyltransferases, carbohydrate transporters—proteins responsible for the movement of carbohydrates across cell membranes—and glycosidases—a class of enzymes that catalyze the hydrolysis of glycosidic bonds—fine-tunes the glycan structures observed on individual proteins and regulates glycoprotein function. Together, these enzymes initiate or elongate glycans, shaping the glycan structures observed on individual proteins.

The resulting glycoproteins display diverse glycoforms, featuring variations in glycosylation site occupancy and the assembled glycan structure [11,12,13,14]. Consequently, the glycoproteome is shaped not only by the glycoproteins themselves but also by macroheterogeneity, characterized by variations in glycosylation site occupancy (i.e., structural diversity influenced by the presence or absence of glycans at specific glycosylation sites) and microheterogeneity, which involves variations in glycan structure (i.e., structural diversity reflecting different glycosylation patterns at individual glycosylation sites) within glycoproteins. For instance, N-glycans have a common pentasaccharide core (Man3-GlcNAc2) with variable distal compositions categorized into three subtypes: high mannose, hybrid, and complex types. Complex structures can be further classified as bi-, tri-, and tetra-antennary, signifying the number of carbohydrate branches originating from the trimannosyl core [11,12,13,14].

It is also crucial to note that the glycome is highly responsive to the dynamic cellular environment and shaped by intra- and extra-cellular conditions, making it context-dependent [15]. The assembly of glycans and the formation of glycoconjugates are governed by various factors, encompassing the accessibility of glycans and cofactors, along with the glycosyltransferases and glycosidases essential for catalyzing such reactions. Additionally, the proper location of each of these elements within the cell and secretory apparatus is crucial. For instance, the HIV-1 envelope protein and the SARS-CoV-2 spike glycoprotein exhibit increased N-glycan processing when present on native virions compared to when these viral proteins are individually expressed in cell lines [16].

Equally important is the recognition that different cells, tissues, organs, and organisms have distinct glycomes and varying vulnerabilities to changes in glycosylation mechanisms [9,17]. While mice lacking ST3GAL5—a gene encoding a protein catalyzing the formation of ganglioside GM3—seem to exhibit only moderate phenotypes, humans with similar defects suffer from severe multisystem disease, such as the Amish infantile epilepsy syndrome, an autosomal, recessive, infantile-onset symptomatic epilepsy associated with developmental stagnation and blindness [4]. Mice inherently express glycan epitopes that are not typically observed in humans, including α-Gal and Neu5Gc (N-Glycolylneuraminic acid, a type of sialic acid) epitopes [18,19]. Similarly, defects in glycosylation pathways that manifest as limited phenotypes in cultured cells can yield severe consequences in intact organisms, even in the form of hypomorphic alleles in humans [4]. For instance, the absence of MGAT1—a gene encoding for a transferase enzyme essential for the synthesis of hybrid and complex N-glycans—results in embryonic lethality in mice but has minimal phenotypic effects in *Arabidopsis* plants and restricted phenotypes in *Drosophila* [4]. In addition to highlighting the complexity of the glycome, these instances underscore the imperative to investigate the impact of glycosylation in native biological states.

To unravel the intricacy of the glycome, there has been a notable surge in analytical methods [20,21,22,23]. These approaches enable mapping glycosylation events on a cellular, tissue, and organism scale, providing insights into their functional roles in biological processes. Resources and repositories have been created to catalogue such diversity [24,25,26,27]. They comprise the Glycosciences.DB (containing data along with nuclear magnetic resonance (NMR) spectra, 3D structures, and analytical tools) [28], UniCarbKB (containing eukaryotic glycans supplemented with NMR, mass spectrometry (MS), and high-pressure liquid chromatography, HPLC data) [24], KEGG Glycan (providing glycan-related data from the Kyoto Encyclopedia of Genes and Genomes) [29], the Japan Consortium for Glycobiology and Glycotechnology (comprising a collection of databases on glycoproteins and glycome-associated diseases supplemented with analytical data) [30], and CSDB (the Carbohydrate Structure Database, which contains structural, taxonomical, bibliographical, NMR spectroscopic, and other data on glycans of bacterial, fungal, and plant origins) [31]. The N-GlycositeAtlas database contains more than 30,000 glycosite-containing peptides (representing > 14,000 N-glycosylation sites) from more than 7200 N-glycoproteins from different biological sources, including human-derived tissues, body fluids, and cell lines from over 100 studies [32]. The field of glycoinformatics, which is relatively new, has emerged to offer scientists diverse tools for accessing, processing, and handling all sorts of carbohydrate-related data. The broad usage of glycomic databases and associated software tools has recently been reported [9,33,34].

These combined studies have begun to decipher the extent of protein glycosylation in health and disease, highlighting the large proportion of proteins undergoing glycosylation. It is estimated that more than 50% of mammalian proteins are glycosylated, emphasizing the prevalence and significance of this modification [35]. The biological processes in which glycosylated proteins are involved are diverse and essential for normal cellular function. For details on how N-linked glycosylation plays an essential role in the folding and the quality control of proteins in the endoplasmic reticulum (ER), we refer the reader to excellent papers [36,37,38]. Beyond these functions, key biological processes in which glycosylation plays a crucial role are cellular recognition and signaling, immune cell recognition, antigen presentation, morphogenesis, the regulation of cell growth, and proper neuronal development and function, processes important for extracellular matrix structure and integrity [4,20,39,40,41,42]. It thus comes as no surprise that defects in protein glycosylation are associated with a variety of diseases. We direct the reader to excellent reviews on glycosylation in cancer [43,44,45,46,47,48], neurodegenerative disorders, including Alzheimer’s disease, Parkinson’s disease, autism spectrum disorder, and schizophrenia [49,50,51], congenital disorders [52,53], infection, and inflammation [54,55,56,57,58]. To understand how the nature and conformation of the glycan can drastically change the interaction of a protein with another in the context of health and disease, we direct the reader to several review articles [47,59,60,61]. Among the well-studied and understood examples are antibodies, where changes in glycan composition at specific sites are known to influence the antigen-binding affinity of monoclonal antibodies, with fucosylation reducing this interaction [62,63,64,65,66]. Another notable example is the cancer cell glycocalyx—a layer of multifunctional glycans that covers the cell surface—where abundant glycosylation, including sialylation, fucosylation, O-glycan truncation, and N- and O-linked glycan branching, has an impact on cell adhesion and the promotion of cancer migration and invasion [18,67,68].

In this review, we delve into the intricate interplay between glycosylation and protein conformation, emphasizing the profound impact of N-glycans on proteins in various diseases. What sets this exploration apart is our focused examination of proteins wherein the presence or absence of specific glycan chains directly shapes the distribution of protein conformations on the energy landscape, dictating the mechanisms of interconversion among these conformations. Importantly, we highlight the role of protein conformational changes and their resulting pathologic functions in understanding the impact of N-glycans. These changes are intricately linked to how the affected protein assembles and interacts with other proteins. This unique focus allows us to unravel the connections between glycosylation and aberrant protein structures, shedding light on the therapeutic implications of targeting conformational and assembly mutants associated with disease. Unlike previous reviews that broadly discuss the role of glycosylation in diseases, our exploration navigates the landscape where the glycome intersects with protein conformation. By doing so, we aim to provide distinctive insights into potential therapeutic avenues and a deeper understanding of the molecular intricacies of how N-glycans impact disease conditions.

## 2. Glycosylation and Protein Conformation

The attachment of N-glycans to specific asparagine residues in a protein’s polypeptide chain introduces bulky and flexible carbohydrate moieties, which can have a dramatic effect on a protein, both locally and distally. For example, Lee et al. performed computational analyses and atomistic molecular dynamics (MD) simulations of six glycosylated and deglycosylated protein pairs sourced from the Protein Data Bank [69]. They found that residues most impacted upon glycosylation were not necessarily near the glycosylated sites, implying that the impact of glycosylation is not localized but rather allosterically propagated to other regions of the protein. Overall, the impact was a decrease in the dynamic movements of the proteins under investigation.

While the study provides valuable insights, a potential limitation arises from the current PDB glycoprotein library. Published crystal structures of naturally occurring glycoproteins tend to be derived from proteins that either had their glycosylation sites mutated or had their glycans partially or completely degraded prior to crystallization or instances where the protein was generated using non-native protein expression systems. Recent advancements, such as the limited deglycosylation assay developed by Coutinho et al., address this limitation by offering a systems-wide approach to analyzing glycoprotein conformational changes upon exposure of cells to a stressor [70]. This method involves native protein-level N-deglycosylation, trypsin digestion, glycopeptide enrichment, peptide-level N-deglycosylation, and quantitative mass spectrometry-based analysis. Applied to LLC-MK2 epithelial cells exposed to dithiothreitol to induce ER stress without impacting protein abundance, the study identified 1145 deglycopeptides. Of these, 1001 peptides, corresponding to 433 unique source glycoproteins, had one or more N-glycosylation sequence motifs. The majority were proteins associated with cellular membranes and were annotated to molecular functions, such as binding of cell adhesion molecules, transmembrane transporters, and glycosyltransferase activity. Their known participation in biological processes was related to cell adhesion, integrin-mediated cell signaling, glycosylation, secretion, and vesicle-mediated transport as biological processes.

Notably, the study revealed that N-glycosites undergoing the highest conformational changes under such minor/moderate ER stress induction were located on loops. Protein loops are inherently flexible regions that readily undergo conformational changes compared to structured elements. The observation that N-glycosites on loops exhibit the highest conformational changes suggests their sensitivity and responsiveness to environmental cues, such as the cellular stress environment. Indeed, loops, along with disordered regions, have recently emerged as crucial connectors for relaying information among structured regions of proteins, including secondary structures and domains, thereby shaping protein structural and dynamic profiles and playing key roles in allosteric communication [71]. Glycosylation of these regions can be expected to have a profound effect on the functional motions of the modified proteins. This functional adaptation may involve proteins undergoing conformational changes as part of their response to changing cellular conditions. Additionally, such changes could influence interactions with other molecules, cellular localization, or participation in signaling or other protein pathways. In summary, the dynamic interplay between glycosylation and protein conformation extends beyond the immediate glycosylation sites, influencing protein dynamics and adding an additional layer of complexity to the mechanisms of functional adaptations.

## 3. N-Glycans’ Effect on Pathologic Protein Conformations in Disease

Considering the allosteric effects of N-glycans in regulating protein conformation, with potential implications for its assembly and function, it is of no surprise that dysregulated N-glycosylation has been implicated in several disease-associated human proteins. Furthermore, these glycans may play a pivotal role in modulating the conformation of pathogen-associated proteins, influencing their infectivity within human cells. In the upcoming sections, we delve into specific proteins to illustrate both scenarios, highlighting instances where glycosylation facilitates cellular transformation and enhances the infectivity of pathogenic agents (see Section 3.1). Additionally, we examine cases where protein glycosylation transforms a physiological protein into a pathogenic, disease-causing form (see Section 3.2 and Section 3.3). The goal is to unravel the nuances of these post-translational modifications, shedding light on how they contribute to the intricate interplay between protein conformation and disease.

### 3.1. Severe Acute Respiratory Syndrome SARS Proteins

Coronavirus Disease 2019 (COVID-19) is a highly transmissible viral infection caused by severe acute respiratory syndrome coronavirus 2 (SARS-CoV-2). The global impact of COVID-19 has been devastating, resulting in over 6 million deaths worldwide. Initially reported in Wuhan, Hubei Province, China in late December 2019, SARS-CoV-2 swiftly spread across the globe, prompting the World Health Organization to declare it a global pandemic on 11 March 2020. In 2020, COVID-19 ranked as the third leading cause of death in the United States after heart disease and cancer, accounting for approximately 375,000 deaths [72].

Few glycoproteins have captured as much attention or undergone more detailed investigations than the SARS proteins. Studies have unveiled how glycans can function as shields preventing recognition of viral proteins by the immune system [73] and how they may serve as activators for the lectin pathway [74]. Perhaps most crucially, these investigations have provided significant insights into how glycans can modify and regulate protein conformation, dynamics, and interactions, a topic we will briefly explore below.

In terms of structure and phylogeny, SARS-CoV-2 exhibits similarities to both SARS-CoV and Middle East Respiratory Syndrome Coronavirus (MERS-CoV). It comprises four primary structural proteins: spike (S), envelope (E) glycoprotein, nucleocapsid (N), and membrane (M) protein, along with 16 nonstructural proteins and 5–8 accessory proteins [75]. The surface spike glycoprotein, resembling a crown, is a threefold symmetric homo-trimer [76], where each protein contains approximately 1200 residues (Figure 1A). The spike is positioned on the virion’s outer surface. It undergoes cleavage into an amino N-terminal S1 subunit, facilitating virus incorporation into the host cell. The carboxyl C-terminal S2 subunit includes a fusion peptide (FP), a transmembrane domain (TM), and a cytoplasmic domain crucial for virus–cell membrane fusion [77]. The S1 subunit is further divided into a receptor-binding domain (RBD) and an N-terminal domain (NTD), playing roles in viral entry and serving as a potential target for neutralization by antisera or vaccines [78]. The RBD is pivotal in infection pathogenesis as it binds to the human angiotensin-converting enzyme 2 (ACE2) receptors in the respiratory epithelium. Following viral attachment, the spike protein S2 subunit is primed by the host transmembrane serine protease 2, facilitating cell entry and subsequent viral replication [79].

Initiation of the binding process between the trimeric S glycoprotein and human ACE2 is marked by the transition of at least one protomer’s RBD from a ‘down’ (closed) to an ‘up’ (open) state [80,81,82] (Figure 1B). This dynamic conformational change involves the transient interconversion of states through a hinge-like motion exposing the receptor binding motif (RBM), which is composed of RBD residues S438 to Q506 [83]. The RBM is buried in the inter-protomer interface of the down S protein; therefore, binding to ACE2 relies on the stochastic interconversion between the ‘down’ and ‘up’ states. The spike S has 22 predicted N-glycosylation sites per protomer, with at least 17 experimentally demonstrated to be occupied [73,84]. Among these, the S1 subunit features 13 putative N-glycosites (N17, N61, N74, N122, N149, N165, N234, N282, N331, N343, N603, N616, and N657), each with the N-X-S/T (X ≠ P) sequon, and one putative N-glycosite (N334) with the N-X-C (X ≠ P) sequon. The S2 subunit, on the other hand, includes nine putative N-glycosites (N709, N717, N801, N1074, N1098, N1134, N1158, N1173, and N1194), all exhibiting the N-X-S/T (X ≠ P) sequon [16]. Several of these sites play a crucial role in regulating the conformational movements of the protein as well as its dynamics, and, in turn, they have an impact on binding to ACE2 and on infectivity [73,83,85,86,87,88,89,90] (Figure 1B).

MD simulations have revealed detailed information about the structural stability and the role of glycosylation for both the ‘down’ and ‘up’ states, as well as for inter-residue interactions and details of binding to ACE2. A study investigating the SARS-CoV-2 spike protein has emphasized the functional significance of glycans at N165 and N234 in regulating the ‘up’ and ‘down’ conformational states of the spike [73]. Through multiple microsecond-long, all-atom, explicitly solvated MD simulations of the full-length SARS-CoV-2 S glycoprotein with a complete glycosylation profile consistent with glycomic data, the study has unveiled a crucial structural role of N-glycans linked to N165 and N234 in modulating the conformational transitions of the RBD. When the RBD transitioned to the ‘up’ state, the glycan at N234 rotated into the resulting void, stabilizing the ‘up’ conformation. Simulating the deletion of these glycans via N165A and N234A mutations resulted in a destabilizing effect on the RBD, prompting a conformational shift toward the ‘down’ state (i.e., a state unfavorable for ACE2 receptor binding). This altered RBD conformation substantially reduced binding to ACE2, as confirmed by biolayer interferometry experiments. Consequently, the specific N-glycans at positions N165 and N234 have been identified as essential structural elements for maintaining the SARS-CoV-2 spike protein in a conformation conducive to ACE2 recognition, facilitating subsequent viral entry into host cells.

To explore the pathway of the fully glycosylated SARS-CoV-2 spike protein opening its RBD, a study employed the weighted ensemble (WE) path-sampling strategy, allowing for atomistic simulations of the spike-opening process [91]. WE, as a path-sampling strategy, directs computational resources toward the transitions between stable states rather than the stable states themselves [92]. This is achieved by running multiple trajectories concurrently and periodically duplicating trajectories that transition between previously and newly visited regions of configurational space. This minimizes the time spent waiting in the initial stable state for transitions over the free energy barrier. The extensive WE MD simulations of the glycosylated SARS-CoV-2 spike head, characterizing the transition from the ‘down’ to ‘up’ conformation of the RBD, revealed a significant gating role for the glycan at N343. This glycan lifted and stabilized the RBD throughout the opening transition. The study also identified an ‘open’ state of the spike RBD, where the N165 glycan of chain B remained the last contact with the RBD on the route to further opening of S1. In conjunction with prior studies by Casalino et al. [73], the research underscored the crucial role of N343 in gating and facilitating the RBD-opening process, emphasizing the necessity of sampling functional transitions for a comprehensive understanding of mechanistic details.

Pang et al. elucidated two-dimensional free-energy landscapes depicting the opening and closing transitions of the SARS-CoV-2 S protein, considering both glycosylated and un-glycosylated forms [93]. The study emphasized the influence of glycans on each state and their role in modifying the kinetics of spike opening. It introduced a nuanced perspective on the role of glycans, suggesting a more intricate impact than previously recognized, especially regarding the glycans at N165 and N343. According to the research, these glycans may affect both the ‘down’ and ‘up’ states, creating a local energy minimum for each. Specifically, the study proposed that these glycans could wrap around the RBM when the RBD is in the ‘down’ state, effectively maintaining it in that configuration. Consequently, these two glycans at N165 and N343 were identified as contributors to stabilizing both ‘down’ and ‘up’ states, establishing a local energy minimum for each.

Glycans also contribute to infectivity, specifically the fusion peptide’s ability to capture the host cell. In the process of spike-protein-mediated fusion, the fusion peptides need to be released from the protein core and associate with the host membrane. Successful infection depends on the transition between pre-fusion and post-fusion conformations. To mechanistically describe this pre-to-post rearrangement and understand the impact of glycans, a study conducted thousands of simulations using an all-atom model with simplified energetics [94]. These simulations revealed that the steric composition of the glycans can induce a pause during the conformational change of the spike protein. This glycan-induced delay presents a crucial opportunity for fusion peptides to effectively capture the host cell, a process that would be inefficient in the absence of glycans. Therefore, the steric composition of both the spike protein and glycans may guide the overall dynamics of host–membrane capture.

In sum, the SARS-CoV-2 S protein undergoes conformational changes crucial for host cell entry (Figure 1). Key N-glycosylation sites stabilize the ‘up’ conformation, facilitating ACE2 recognition and viral entry. Select N-glycans act in gating and stabilizing the S protein during conformational transitions. Furthermore, N-glycans actively contribute to SARS-CoV-2 infectivity by influencing the dynamics of the fusion peptide. This peptide is instrumental in capturing the host cell during the spike-protein-mediated fusion process. The steric composition of glycans induces strategic pauses in conformational changes, creating vital windows for efficient host cell capture. To sum up, strategically positioned N-glycans intricately regulate the conformation and dynamic movements of the S protein. This regulation, in turn, dictates its ability to attach to the ACE2 receptor, enter the cell, and initiate infection in the host.

### 3.2. Prion Protein

Prion diseases, or transmissible spongiform encephalopathies (TSE), are a class of fatal, infectious neurodegenerative illnesses that affect both humans and animals. Among human prion diseases are Creutzfeldt–Jakob Disease (CJD), Variant Creutzfeldt–Jakob Disease (vCJD), Gerstmann–Straussler–Scheinker Syndrome, Fatal Familial Insomnia, and Kuru. Most prevalent in animals are bovine spongiform encephalopathy (BSE) (or Mad Cow Disease) in cattle, chronic wasting disease (CWD) in deer, elk, and moose, and scrapie in sheep [95,96,97]. These diseases occur when a normally occurring protein, the prion protein, undergoes a pathogenic transformation, where the key factor driving the pathology is a conformational change. Prion proteins, occupying a middle ground between viral and human counterparts, are the subject of our second discussion exploring how N-glycans serve as modulators of protein conformation.

Prion diseases are caused by the conformational rearrangement of the endogenous cellular prion protein (PrP^C^) into an abnormal, toxic form (PrP^Sc^) [98]. PrP^C^, an *N*-glycosylated protein, is tethered to the outer leaflet of the plasma membrane through a glycosyl phosphatidylinositol (GPI) moiety [99,100]. While expressed across various tissues, it is particularly enriched in neurons, making them the primary sites susceptible to TSE-related degeneration [101]. N-Glycosylation influences the conformational stability of PrP [102,103,104,105,106]. Changes in glycosylation patterns can impact the folding kinetics of PrP and may contribute to the transition from the normal, cellular form to the disease-associated form. Different glycoforms may exhibit variations in their susceptibility to conversion into the pathological PrP^Sc^ form, affecting the progression of prion diseases. The impact of the glycans, or the lack of, however, may be strain-dependent, and the conversion of PrP^C^ into PrP^Sc^ may be sustained through several pathways depending on the origin of the disease [102,103,104,105,106]. In the context of infectious TSE, it is proposed that exogenous PrP^Sc^ interacts with PrP^C^, acting as a template for the conversion of the latter into an additional PrP^Sc^ molecule. In familial prion diseases, mutations within the prion protein gene are suggested to facilitate the folding of PrP into a pathogenic conformation [102,107,108,109].

Here, we intend not to delve into this complexity but rather explore how changes in glycosylation may support pathogenic prion protein conformation, interaction, and function. As anticipated for a membrane protein, the newly synthesized 254-amino acid PrP^C^ undergoes cleavage of its hydrophobic N-terminal signal peptide (residues 1–23) to enable its targeting to the rough endoplasmic reticulum, the initial compartment of the secretory pathway. Within this compartment, co- and post-translational modifications of PrP^C^ take place, including the addition of high mannose-type N-glycans (Glc_3_–Man_9_–GlcNAc_2_) at positions Asn181 and Asn197 in human *(hu*Pr^P^) and Syrian hamster (*Sh*Pr^P^) PrP^C^ (corresponding to Asn180 and Asn196 in mouse, *Mo*Pr^P^), formation of a unique disulfide bond, and attachment of a GPI moiety following cleavage of the hydrophobic C-terminal signal peptide. The glycosylation sites of PrP^C^ exhibit variable occupancy, resulting in non-, mono-, or di-glycosylated forms [100,102,103,106]. The N-terminal segment (residues 23–125, after removal of the signal peptide) exhibits significant flexibility. This segment encompasses an octapeptide repeat domain capable of binding divalent ions, such as Cu^2+^ and Zn^2+^. In contrast, the C-terminal domain (residues 126–230), housing the glycan attachment sites, adopts a distinctive structure characterized by three α-helices, labeled as helices one through three (one through three or A through C) and two anti-parallel β-strands flanking helix 1 (Figure 2A).

In cellulo and in vivo studies support the role of these glycans in modulating a pathologic conformation and function of PrP [102,110,111,112] (Figure 2B). For example, since *Sh*PrP contains complex-type oligosaccharides attached to Asn residues 181 and 197 (as in *Hu*PrP), an early study mutated the Thr residues to Ala within the NXT consensus sites [113]. This substitution disrupts the specific motif required for N-glycosylation, hindering the proper attachment of glycan moieties to the protein. Single and double glycosylation site mutations were expressed in transgenic mice deficient for mouse *Mo*PrP, and the brains were analyzed for the distribution of mutant *Sh*PrP^C^. The analysis focused on the hippocampal region in each case. Wild-type *Sh*PrP^C^ was predominantly found in the dendritic trees of the CA1 to CA4 regions of Ammon’s horn and the dentate gyrus. It was notably absent from the cell bodies of pyramidal and granule cell layers in these regions and largely absent from white matter tracts like the corpus callosum. In contrast, mutation in either one or both glycosylation consensus sites had a significant impact on the anatomical distribution of *Sh*PrP^C^. Mutation of the first glycosylation site alone or in combination with the second site led to low levels of mutated *Sh*PrP^C^, its accumulation in nerve cell bodies, and limited presence in dendritic trees. When the second glycosylation site was mutated, *Sh*PrP^C^ levels were comparable to wild-type *Sh*PrP^C^. This mutant protein, however, was observed throughout all neuronal compartments, including the cell body, dendritic tree, and axons in the white matter. Transgenic mice with inactivation at the second Asn197 site (T199A) supported prion replication upon infection, while mice mutated at the first site appeared resistant [99,113].

A recent study employed knock-in mouse models expressing cell surface PrP^C^ with zero or two N-glycans and several complementary approaches to address the impact of glycosylation on prion protein localization and function [106]. Mice expressing PrP^C^ without glycosylation were generated through the introduction of two-point mutations at the endogenous *Prnp* locus using a single guide RNA. These mutations, corresponding to the substitution of asparagine to glutamine at positions 180 and 196 (in accordance with mouse PrP numbering), led to alterations in the N-glycosylation sequons. PrP(180Q196Q)) mice exhibited normal expression and trafficking of PrP^C^ with no evidence of spontaneous prion disease. However, a significant difference in susceptibility to prion infection was observed between PrP(180Q196Q) mice and wild-type mice. Notably, the PrP(180Q196Q) mice consistently showed more severe spongiform degeneration across all strains compared to wild-type mice. Additionally, upon prion infection, these mice displayed marked atrophy of the hippocampus due to severe neuronal loss, including complete loss of CA1 pyramidal neurons and the presence of numerous gemistocytic astrocytes, reflecting the enlarged and filled appearance of the cell. Gemistocytic astrocytes are often associated with certain pathological conditions. This effect persisted upon second passage. In contrast, wild-type mice exhibited moderate loss of hippocampal neurons. Furthermore, the cerebellum of all infected PrP(180Q196Q) mice lacked PrP^Sc^, a notable distinction from wild-type mice where all three strains were present in the cerebellum. Importantly, the absence of PrP^Sc^ in the cerebellum of PrP(180Q196Q) mice was not attributed to a lack of PrP^C^ expression, as PrP(180Q196Q) was expressed in the cerebellum at levels similar to wild-type PrP. The morphology of PrP^Sc^ in PrP(180Q196Q) brains differed from wild-type mice, as most PrP(180Q196Q) brains displayed either plaque-like deposits or dense parenchymal plaques. Consequently, PrP(180Q196Q) mice when exposed to subfibrillar prion strains manifested distinct disease characteristics, including an elevated presence of plaques and plaque-like structures, more pronounced cortical spongiosis, and a conspicuous absence of prions in the cerebellum.

A study by Yi and colleagues [111] explored the impact of glycosylation on various aspects of PrP using cultured cells expressing wild-type PrP and glycosylation mutants. The study found that glycosylation significantly influenced the subcellular localization, resistance to proteolytic digestion, and aggregation ability of human PrP. Wild-type PrP and monoglycosylated mutants—N181D, N197D, and T199N/N181D/N197D—were primarily attached to the plasma membrane, while pathological mutants with altered glycosylation sites (i.e., PrP F198S) and unglycosylated PrP mutants (i.e., N181D/N197D) were mainly present in the cytoplasm. Furthermore, the study revealed that the degree of glycosylation correlated with the protein’s proteolytic resistance and aggregation ability. PrP with fewer glycosylation modifications exhibited higher aggregation propensity and a higher degree of abnormal conformers, as measured by its resistance to protease digestion. Additionally, glycosylation deficiency increased the vulnerability of the protein to stressors and enhanced its cytotoxicity.

In the context of understanding the impact of N-glycans, it is crucial to underscore the role of protein conformational changes and their resulting pathologic functions, which are intricately linked to how the affected protein assembles and interacts with other proteins. Glycosaminoglycans, especially heparan sulfate and heparin, are considered crucial molecules for prion conversion and infection. For instance, in prion diseases, heparan sulfate, a prevalent polyanion in the brain, accelerates disease progression by facilitating the conversion and assembly of extracellular, ADAM10-cleaved PrP into parenchymal plaques [114]. In mice, di-glycosylated PrP^Sc^ demonstrated the least heparan sulfate binding, while unglycosylated PrP^Sc^ exhibited the highest heparan sulfate binding [106]. Similarly, glycans also disrupt PrP binding to heparin, with di-glycosylated PrP^C^ exhibiting the lowest affinity for heparin binding [102]. Notably, PrP^C^ with two to three glycans displayed low heparin affinity, while unglycosylated PrP^C^ showed high affinity, with this affinity progressively decreasing with each additional glycan. The heparin-binding affinity of PrP^C^ from age-matched wild-type and PrP(180Q196Q) mouse brain homogenates followed a similar pattern—unglycosylated PrP^C^ exhibited higher heparin-binding affinity than glycosylated wild-type PrP^C^, a trend also observed in their ADAM10-cleaved counterparts [106].

Structural and biochemical studies have provided explanations for how these glycans may affect the prion protein. From early NMR structures of the full-length and N-terminally-truncated forms of recombinant *Mo*PrP, *Sh*PrP, and *Hu*PrP, it became known that the entire N-terminal segment (residues 23–126) is flexibly disordered and that only the C-terminal part (residues 127–231) possesses a defined 3D structure [115,116]. MD simulations on the C-terminal region of *Hu*PrP (residues 90–230), with and without the glycans, suggested the structured part of the *Hu*PrP protein (residues 127–227) was stabilized overall from addition of the glycans, specifically by extensions of Helix-B (i.e., helix 2) and Helix-C (i.e., helix 3) and reduced flexibility of the linking turn containing Asn197. The stabilization appeared indirect and not from specific interactions, such as H bonds or ion pairs. Thus, glycosylation at Asn197 has an allosteric effect, with an impact on stabilizing a conformation of the protein [117].

Recent biochemical studies have provided additional insights into the role of glycans in the prion protein. NMR and electron paramagnetic resonance spectroscopy studies suggest that the two N-glycans play a crucial role in maintaining an intramolecular interaction, effectively bringing together the C- and the N-terminal domains, as elaborated further [112,118,119]. This interaction is mediated through a Cu^2+^–histidine tether, bringing the C- and N-terminal domains into proximity and likely stabilizing the overall structure while reducing dynamic motion (Figure 2C). Additionally, PrP glycans at N181 and N197 actively promote the interaction between the N-terminal and C-terminal domains, synergizing with the effect of His-Cu^2+^ coordination [112,120]. A patch of negatively charged amino acids on the same protein surface as the histidines and glycans serves as a third contributor to these interactions. These interactions play a functional role in suppressing the neurotoxic activity of PrP^C^, as demonstrated by studies on the PrP mutant N180Q/N196Q. This mutant, where Asn residues at the glycan attachment sites were replaced with Gln residues to prevent glycosylation while preserving the polar carboxamide side chain common to both Asn and Gln, exhibited effects similar to a highly toxic deletion mutant of PrP [112,120].

Together, these studies indicate that the loss of glycans destabilizes the prion protein structure, enriching for a conformation that enables pathologic interactions. Loss of glycosylation could increase the affinity of PrP^C^ for a particular conformer of PrP^Sc^ and of other pathologic interactors (such as heparan sulfate) determining the rate of nascent PrP^Sc^ formation and the specific patterns of PrP^Sc^ deposition (Figure 2). Intriguingly, distinct brain regions, and presumably cell types, demonstrated distinct vulnerability to these pathologic conformers. Also, the lack of glycans increased the vulnerability of the protein to additional stressors, increasing its pathogenicity.

### 3.3. Glucose-Regulated Protein 94 (GRP94)

Glucose-regulated protein 94 (GRP94), also known as endoplasmin and gp96, serves as a crucial molecular chaperone located in the ER of eukaryotic cells [121]. Its primary functions involve ensuring proper folding, maturation, and quality control of client proteins within the ER. Beyond its role in protein folding, GRP94 actively participates in various cellular processes, contributing to cellular homeostasis and overall cell function [122,123]. GRP94 forms a homodimer, and each chain comprises three domains: the N-terminal (NTD), middle, and C-terminal (CTD) (Figure 3A). Mechanistically, the chaperone undergoes ATP-driven conformational changes associated with the folding of a client protein. ATP binding at the NTD induces a shift in the chaperone to a closed conformation, inducing changes in regions critical for protein client binding. Following ATP hydrolysis, rearrangements occur in the residues of the client-binding site, leading to mechanical translation into conformational changes in the bound client [122,123].

While primarily localized to the ER, GRP94 is also found in the cytosol, at the cell surface, and extracellularly [122,124,125]. This altered distribution is often associated with and intensified in disease-related scenarios [126,127] (Figure 3B). For instance, pathogens utilize surface GRP94 to infect host cells [128,129,130]. In autoimmune diseases, overexpression of cell-surface GRP94 enhances toll-like receptor function and downstream signaling through MyD88 [131]. In cancer cells, surface GRP94 imparts an aggressive phenotype by regulating the stability of receptor tyrosine kinases (RTKs), such as HER2 and EGFR, inhibiting their internalization and enhancing their aberrant downstream signaling [132,133,134,135]. In inflammatory diseases, surface GRP94 induced a pro-inflammatory profile in macrophages [136,137]. Similarly, an extracellular GRP94 complexed with immunoglobulin Gs (IgGs) contributes to the pathogenesis of type 1 diabetes [138].

N-glycosylation plays an important role in the generation of such pathologic surface-expressed and extracellular GRP94 forms (Figure 3C). GRP94 contains six potential *N*-glycan acceptor sites, namely, N62, N107, N217, N445, N481, and N502, yet under normal conditions, the protein is predominantly monoglycosylated at N217 [139,140]. Cherepanova and colleagues [139] demonstrated that GRP94 can become glycosylated at all sites in cells that are exposed to oligosaccharyltransferase inhibitors or low doses of ER stress-inducing agents and in cells with partial or complete loss of oligosaccharyltransferase complex activity. In a later study, Wen and colleagues demonstrated that glycosylation of silent sites was heterogenous. Some sites, such as N62, were enriched with mannosylated N-glycans, such as Man_9_GlcNAc_2_ and Glc_1_Man_9_GlcNAc_2_ (N2H9 and N2H10), whereas N107 and N445 had a varied content of mannosylated N-glycans but also fucosylated or sialofucosylated complex-type N-glycan structures [141].

The regulation of GRP94 glycosylation, promoting the consistent omission of silent sites in non-stressed cells and facilitating swift and efficient glycosylation of silent sites in stressed cells, poses intriguing questions. None of the GRP94 sequons have a negative flanking sequence score (i.e., surrounding amino acid sequences do not hinder glycan addition), indicating that these sites are not skipped due to suboptimal conditions. Cherepanova et al. [139] proposed instead that the existence of a mechanism restricting nascent GRP94 access to the STT3A active site, blocking cotranslational glycosylation of the silent sites, is responsible for the lack of glycosylation on the silent sites. In this proposed scenario, the N62 and N107 sites in GRP94 could enter the STT3A active site before the normal glycosylation site (N217) is incorporated into the nascent chain. However, factors associated with cellular stress may saturate pathways responsible for blocking glycosylation of the silent sites in GRP94 by oligosaccharyltransferase. Consequently, the N62 and N107 silent sites could become glycosylated by oligosaccharyltransferase complexes in cells exposed to stressors, potentially contributing to disease-related glycosylation patterns. One cannot exclude, however, that other (probably context-specific) alterations in the glycosylation machinery may exist that shape the occupancy of individual silent sites, with each possibly impacting the conformation and function of the GRP94 protein.

Supportive of this notion, N-glycan occupancy at these silent sites was indeed reported in disease. In OVCAR-3 ovarian cancer cells, which exhibit high levels of EGFR at the plasma membrane [142], a study employed an unbiased, large-scale MS method to determine N-glycosylation site occupancy by comparing tunicamycin treated to inhibit the overall N-glycosylation occupancies of the cells and untreated cells [143]. Using this method, they determined that GRP94 had four of the sites—N62, N107, N217, and N481—occupied in OVCAR-3 cells. Out of these sites, N217 had a high occupancy (~100%) and N62 occupancy was moderate (~50%), followed by N107 (<50%) and N481 (minor, ~10%). Yan and colleagues performed glycosylation site mapping through mass spectrometry and identified N62, N217, and N502 as putative N-glycosylated sites on a GRP94 variant enriched on the cell surface of MDA-MB-468 cells, an EGFR-overexpressing triple negative breast cancer cell line [134]. Here, too, N217 was a high-occupancy site, with N62 and N502 being partially occupied.

Biochemical, functional, and structural investigations unequivocally affirm the pathological implications of N-glycan occupancy at specific silent sites [130,134,136,137,144]. Within certain breast cancer cell subtypes, the glycosylation event at N62 emerges as a pivotal factor contributing to their aggressive phenotype, resistance to therapy, and immune evasion [134,136,145]. This transformation leads to the stabilization of a unique conformation of GRP94, with significant repercussions [134,144]. Primarily, N-glycosylation at N62 serves as a structural mediator, inducing the conversion of the GRP94 chaperone into epichaperomes—hetero-oligomeric forms tightly composed of chaperones, co-chaperones, and other factors [134,146,147]. In these epichaperomes, GRP94 adopts scaffolding functions not observed in normal cells, where GRP94 primarily participates in protein control and folding. Through this scaffolding function, GRP94 influences the assembly and connectivity of proteins crucial for maintaining a malignant phenotype, enhancing their activity. Consequently, the functions of these proteins are markedly enhanced, leading to the aberrant remodeling of dependent cellular protein networks. This provides a survival advantage to cancer cells and tumor-supporting cells within the tumor microenvironment. While the precise composition of the glycan at N62 remains unknown, evidence suggests that the modified residue, rather than the specific sugar structure, is crucial for mediating the cancer-supporting conformation of GRP94 in the N62 glycoform [134,144,148]. Despite this uncertainty, its susceptibility to deglycosylation by EndoH suggests a high mannose glycan characteristic [134]. This notion is also supported by glycoproteomics studies from Wen and colleagues, which found N62 to be enriched in mannosylated N-glycans, such as Man_9_GlcNAc_2_ and Glc_1_Man_9_GlcNAc_2_ (N2H9 and N2H10) [141].

Among the proteins affected by this pathologic GRP94 glycoform are RTKs. In breast tumors characterized by the overexpression of RTKs like HER2 and EGFR, GRP94 with the N62 occupied site becomes enriched at the cell surface [134]. This enrichment plays a pivotal role in reducing RTK internalization and maintaining the RTK in a state conducive to constitutively enhanced downstream signaling. To validate these findings, Yan and colleagues utilized CRISPR-Cas9 to manipulate endogenous GRP94, generating homozygous clones—N62Q, N217A, and N62Q/N217A—from the MDA-MB-468 breast cancer cell line [134]. Clones expressing GRP94(N62Q) exhibited a significant absence of EGFR at the plasma membrane, accompanied by a lack of EGFR-signaling activity. Conversely, the GRP94(N217A) mutants, representing the folding form of GRP94, demonstrated no deleterious effects on EGFR levels or signaling [134]. This underscores the pathologic gain-of-function nature of N62 glycosylated GRP94 in the context of cancer. Specifically for GRP94, at the plasma membrane (where ^Glyc62^GRP94 is located), ^Glyc62^GRP94 may promote cancer as, by forming epichaperome platforms [134,146,148], it provides a backbone upon which oncogenic proteins and protein assemblies cluster, augmenting their pathologic function and leading to an aggressive phenotype in ^Glyc62^GRP94-expressing cancer cells (Figure 3B).

To understand how N-glycans at N62 and N217 influence GRP94 conformation and dynamics, Castelli et al. employed MD simulation studies [144]. Their study revealed dynamic modulation of GRP94’s conformation and interactions by these glycans, impacting protein interaction mode and ATP processing essential for folding (Figure 3C). In the fully glycosylated state, sugars obstructed the N-terminal domain (NTD), impeding ATP binding. The N62 glycan favored an open, ATPase-incompetent NTD lid conformation, influencing the charged linker between the NTD and M-Domain, favoring a more closed and inaccessible protein client binding site that is thus unfavorable for folding. Therefore, the N62 glycan-induced alterations in conformational ensembles significantly impact the efficiency of translating nucleotide-encoded signals. Conversely, the N217 glycan had little impact on these factors. The results suggest that N62 glycosylation actively shifts GRP94 from a foldase to a protein-assembly platform, impacting the ATP-binding site’s efficiency and perturbing the charged linker’s dynamics. Ultimately, glycosylation induces GRP94 malfunction, disrupting its structural ensembles and chaperone cycle kinetics and leading to interactome remodeling at a much larger scale than the simple local covalent modification that might lead to hypothesize, amplifying dysfunction, and remodeling cellular phenotypes.

In summary, the N-glycosylation of silent sites in GRP94 during disease profoundly influences the protein’s conformation, assembly, and function (Figure 3). Under normal physiological conditions, GRP94 is glycosylated at N217 and localized to the ER, and it facilitates client protein folding through transient interactions (Figure 3A–C). However, N-glycosylation at N62 disrupts GRP94’s ER confinement, leading to a conformational shift that promotes stable interactions with proteins at the plasma membrane, enhancing their functions and inducing aberrant remodeling of cellular protein pathways. This glycosylation transforms GRP94 from a folding chaperone to a scaffolding protein, consequently reshaping protein assembly and connectivity, resulting in systems-level dysfunction. Consequently, alterations in N-glycosylation at N62 generate a distinct protein with unique conformational, dynamic, and functional characteristics compared to normal GRP94 in healthy cells. Thus, GRP94 is a unique example where N-glycosylation increases the pathologic properties of a protein indirectly by modulating its complexation.

## 4. Therapeutic Implications—Targeting Conformational and Assembly Mutants in Disease

Understanding the pivotal role of N-glycans in shaping the conformation and assembly of these proteins is not only crucial for unraveling the intricate mechanisms underlying disease pathological mechanisms but also presents unique opportunities for therapeutic intervention (Figure 4). Disease-specific conformations, regulated by glycosylation, are conformational mutants and as such provide a unique target for intervention. A conformational mutant refers to a mutant form of a protein that differs from the wild-type (normal) protein in its three-dimensional structure or conformational dynamics. Thus, the conformational and assembly mutants induced by aberrant N-glycosylation in these proteins serve as actionable targets, offering avenues for targeted interventions through antibodies, small molecules, and other innovative approaches. This exploration opens new frontiers in the development of precision therapies, highlighting the potential to modulate disease-associated protein structures and functions with a high degree of specificity. We discuss how the intricate interplay between N-glycans and protein conformational dynamics paves the way for novel therapeutic strategies.

### 4.1. Targeting the Viral Protein Conformations

In the preceding sections, we extensively explored the mechanisms underlying SARS-CoV-2 infectivity, unraveling crucial insights that have paved the way for therapeutic advancements. Leveraging this knowledge, therapeutic strategies have been devised to manipulate glycan-mediated conformational changes in the viral spike protein, aiming to attenuate infectivity and elicit a robust immune response targeting diverse epitopes. Notable contributions include the development of mRNA vaccines encoding stabilized soluble S trimers in the prefusion conformation, fostering neutralizing antibody responses for robust host protection against viral infection [149].

Positioned on the virion’s outer surface, the spike protein undergoes cleavage into S1 and S2 subunits, where the absence of N-glycans at positions N165 and N234 on S1 significantly impacts the conformational flexibility of the RBD. These glycans play a pivotal role in stabilizing the RBD in the ‘up’ conformation, which is crucial for effective ACE2 receptor binding and viral invasion [77] (Figure 1). Casalino and colleagues [73] proposed that the introduction of N165A and N234A mutations may provide a means to deliberately control the RBD’s conformational dynamics, favoring a predominantly ‘down’ state. This alteration could potentially yield less infectious viruses with RBDs shifted toward the closed state, making the virus more vulnerable to immune detection. An alternative approach, presented by Dodero-Rojas et al., targets the conformational change in S2 responsible for membrane fusion [94]. This highlights the potential to impede viral entry by targeting intermediates of the spike protein’s conformational change.

The critical structural role of specific N-glycans in modulating the SARS-CoV-2 spike protein’s dynamics and conformational states has also informed the rationalization of neutralizing antibody activity. With over ten thousand neutralizing antibodies discovered, the primary target is the RBD, although some also target the NTD and other non-RBD epitopes [150,151,152]. A study by Pang et al. [93] investigated the differential impact of glycans on epitope exposure, considering the spike-opening dynamics of the SARS-CoV-2 S protein and the influence on various antibodies. The analysis by Pang and colleagues involved seven antibodies with epitopes covering RBD and NTD regions. They found that epitope exposure, measured by accessible surface area, was observed to remain constant or increase significantly during the transition from the ‘down’ to the ‘up’ state of RBD, impacting the binding of some but not all of the evaluated antibodies. Other studies used explicit dynamic information on wild-type SARS-CoV-2 S protein and several mutants to reveal the intrinsic determinants of epitope presentation and the mechanisms of antibody escape [153,154,155]; in all of these cases, mutations modify the glycan-shield of the S protein with a strong impact on its dynamics and recognition properties.

### 4.2. Correcting the Prion Protein Conformation

Numerous investigations have highlighted that the pivotal event in the pathogenesis of prion diseases is the conformational transition of prion protein (PrP) from its normal cellular form (PrP^C^) into an abnormal form (PrP^Sc^). While the precise mechanisms governing the transition from PrP^C^ to PrP^Sc^ and the structure of PrP^Sc^ remain unknown, N-glycans play an important role, as detailed in prior sections above. In Papua New Guinea, a genotype, V127M129, has been identified as completely resistant to prion diseases [156,157]. To decipher the disease-resistant effect of the G127V mutant, Zheng and colleagues [158] conducted detailed structural and dynamic analyses on the G127V-mutated human PrP (residues 91–231) in comparison with wild-type PrP. Their investigations revealed distinctive alterations induced by the G127V mutation. The G127V variant exhibited a stretch-strand (SS) pattern comprising two segments (SS1: Y128-G131; SS2: V161-R164), while the wild-type protein displayed a well-defined β-sheet with two strands (β1: Y128-G131; β2: V161-R164). The larger hydrophobic side chain of Val127 introduced steric hindrance, leading to a significant rearrangement in the side chain of Tyr128. Consequently, the study suggests that interventions hindering the conversion of SS1 (Y128-G131) and SS2 (V161-R164) segments and adjacent regions into a stable β-sheet could potentially thwart the pathological conformational transition of PrP. The study therefore underscores the potential for therapeutic development by correcting specific pathologic PrP conformations, leveraging an understanding of structural differences between the wild-type and the G127V protein.

Others have opted for designing small molecules, termed medical chaperones, that could specifically bind to and stabilize the native conformation of PrP and prevent abnormal aggregate formation [159]. Along these lines, Yamaguchi and collaborators developed a small molecule, termed MC, specifically designed to stabilize the normal conformation, PrP^C^ [160]. Given that the normal conformation, PrP^C^, is highly conserved across species [161], this approach holds promise as it may be effective across various prion diseases and independent of strain or species. MC demonstrated the ability to stabilize the normal cellular prion protein, eliminate prions in infected cells, prevent the formation of drug-resistant strains, and directly inhibit the interaction between prions and abnormal aggregates [160]. In prion-infected mice, MC extended their survival, while in macaques infected with bovine transmissible spongiform encephalopathy, MC slowed down the development of neurological and psychological symptoms and reduced the concentration of disease-associated biomarkers in cerebrospinal fluid [160].

### 4.3. Targeting Pathologic GRP94 Conformers and Assemblies

GRP94 has emerged as a key player in cancer [124,162,163]; however, clinical therapeutics targeting the protein have yet to be developed. This is largely attributed to our insufficient understanding of how to differentiate GRP94 in normal cells from that expressed by tumors and tumor-supporting cells. The identification of glycosylation on N62 of GRP94 as a mechanism for the formation of conformational mutants associated with protein assembly mutations in cancer [134,144] opens new avenues for developing therapeutics targeting disease-associated GRP94 forms. From a therapeutic perspective, the focus is on understanding the impact of N62 glycosylation on GRP94 conformation and assembly, influencing function, rather than identifying defects in the glycosylation machinery leading to impaired glycan processing of GRP94.

Yan and colleagues have shown that tumor ^Glyc62^GRP94 (i.e., GRP94 glycosylated at N62) can be selectively targeted over the physiologic GRP94 through therapeutics, such as PU-WS13 [134]. PU-WS13 is a small molecule with preferential activity for ^Glyc62^GRP94 over GRP94 [134,144]. PU-WS13 binds to an allosteric pocket of GRP94 that only partly overlaps with the ATP-binding pocket and is not accessible in closely related paralogs of GRP94 [133,164,165,166]. PU-WS13 is inactive or shows reduced activity in Glyc62GRP94-negative but GRP94-positive cells [133,134]. PU-WS13 shows reduced binding to a GRP94(N62Q) mutant when compared to ^Glyc62^GRP94. Mutations at residue N217, the site required for GRP94 folding functions, in the context of an N-glycosylated N62 residue (i.e., in ^Glyc62^GRP94(N217A)) do not interfere with PU-WS13’s binding. Deglycosylation with EndoH, which retains N-acetylglucosamine attached on Asn, reduces PU-WS13’s binding to GRP94, whereas PNGaseF, which removes the N-glycan in its entirety, leaving the unmodified Asn-NH_2_ residue, abolishes PU-WS13’s binding to GRP94. PU-WS13 is toxic to ^Glyc62^GRP94-expressing cancer cells but not to cancer cells with abundant GRP94 or GRP94(N62Q) but no ^Glyc62^GRP94 expression. GRP94(N62Q)-expressing clones and GRP94 KO clones displayed reduced sensitivity to PU-WS13. Conversely, the GRP94(N217A) mutants (i.e., of the folding GRP94) had no deleterious effect on PU-WS13 activity. Combined, biochemical and functional studies confirm that the biological effects observed with PU-WS13 are ^Glyc62^GRP94-specific [130,134,136,144,145].

Castelli et al. [144] employed MD simulations to unravel the dynamic mechanisms of GRP94 glycoforms. They utilized these findings to rationalize the engagement of specific conformational states by PU-WS13, providing crucial insights into the targeting of the disease-specific, aberrantly-glycosylated GRP94 variant by this small molecule. These simulations also offer structural support, elucidating how pathologically-N-glycosylated GRP94 variants can be selectively targeted and influenced.

Several studies have shown that inhibition of ^Glyc62^GRP94 by PU-WS13 was safe and active against cancers in vivo. For example, in nude mice, PU-WS13 suppressed the growth of ^Glyc62^GRP94-dependent breast tumors without affecting GRP94 functions [134]. Even for the long-term treatment regimens that delivered 37 to 62 doses of PU-WS13 to mice over 87 days, no treatment-related toxicities were observed. In a preclinical model of 4T1 breast-tumor-bearing BALB/c mice, PU-WS13 decreased CD206-expressing M2-like macrophages in the tumor microenvironment, reduced tumor growth and collagen content, and increased the recruitment of CD8^+^ cells in the tumor microenvironment [145]. This is of high clinical relevance as M2 macrophages are strongly associated with fast proliferation, poor differentiation, and estrogen receptor-negativity in human primary breast tumors [167].

PU-WS13 also alleviated inflammatory responses in primary macrophages in a model of alcohol-induced liver damage [137], enhanced pneumococcal clearance from lung tissues and ameliorated lung pathology in a mouse model of influenza A virus infection with secondary bacterial pneumonia [130], reduced the pro-inflammatory profile of macrophages in disease models of inflammation [136], and showed antiviral activity against Dengue virus serotypes and Zika virus strains in multiple human cell lines in viral replication models [129], supporting ^Glyc62^GRP94 as targets in several distinct disease states beyond cancers [129,130,136,137].

## 5. Conclusions and Future Directions

In conclusion, the investigation of the intricate interplay between N-glycans, protein conformational dynamics, and disease pathogenesis not only deepens our understanding of pathophysiological mechanisms but also unveils a rich landscape for innovative therapeutic strategies. The discussions herein illuminate the potential for targeted interventions across viral infections, prion diseases, and cancer, showcasing the transformative power of comprehending and manipulating protein conformations. Importantly, while our examples are focused on these specific diseases, the principles elucidated in this exploration hold promise for impacting a myriad of other disorders where conformational mutants play a pathologic role. This profound insight not only opens new avenues for precision therapies but also underscores the importance of a multidisciplinary approach in developing effective strategies against diverse and complex diseases. As we navigate this frontier, the convergence of disciplines holds promise for a future where tailored interventions based on protein conformational nuances become a cornerstone in the treatment of intricate diseases.

Glycoform-specific targeting is a new frontier in several diseases, including cancer and neurodegenerative diseases. It offers a solution to limiting off-target effects by increasing the disease specificity. In cancer, current therapeutic strategies are based on the development of antibodies, which detect cancer-associated glycoforms of therapeutic targets, such as integrins, RTKs, and other glycoproteins, and the discovery of small molecules that interfere with specific glycosylation pathways altered in disease [46,67]. The platform we describe is unique, as it targets the pathologic conformation and assembly of a protein generated by site-specific glycosylation.

By influencing protein conformations and impacting the abundance of specific forms at distinct cellular locations, glycans may significantly alter cellular networks. Distinct conformers can engage in different interactions, thereby remodeling cellular protein–protein interaction networks at a large scale by shifting the assembly of multiprotein complexes required to regulate biomolecular pathways toward protein assemblies with pathologic activities, known as protein assembly mutations. In this context, understanding the interactome—the complement of interacting partners—of the glycomutant is crucial. The engagement of a protein with various binding partners in diverse biological contexts shapes the functional outcomes of these interactions. Depending on the presence of potential interactors, a protein may be assigned to different protein pathways in distinct cellular contexts [146]. Therefore, systems-level investigations in specific disease contexts are essential to provide informed insights into cellular transformation by conformational mutants. Exploring their interactors is anticipated to provide insights not just into the proteins affected by conformational mutants but, more crucially, into the functional consequences of these alterations at the network and cellular levels. This knowledge may pave the way for novel therapeutic strategies targeting disease-specific protein assemblies particularly focusing on assembly mutants, i.e., disease-specific protein–protein interactions, which represent a promising avenue in this evolving landscape.

## Figures and Tables

**Figure 1 biomolecules-14-00282-f001:**
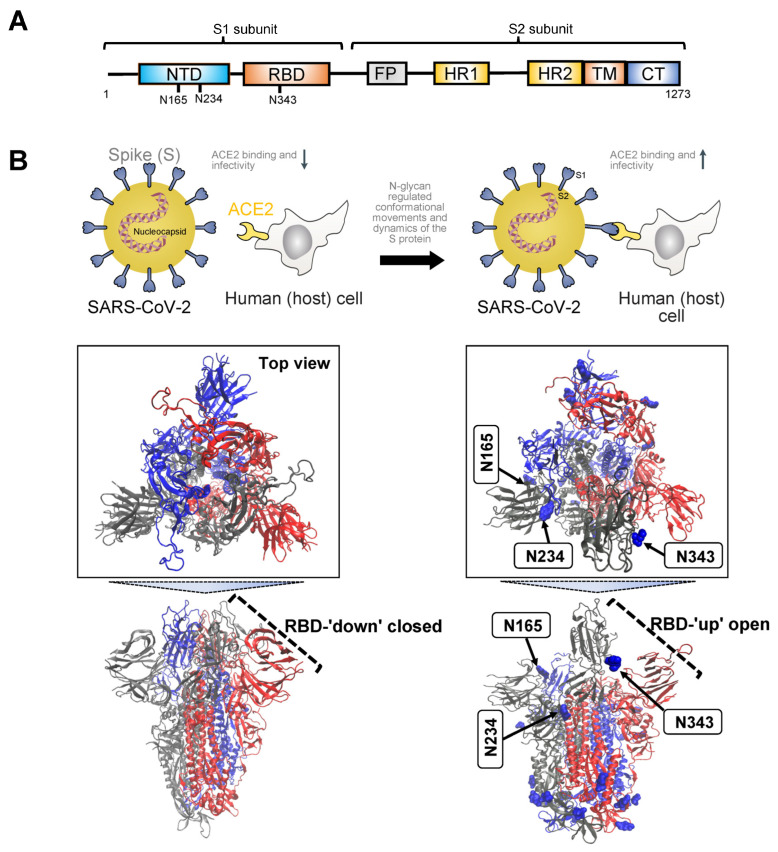
(**A**). Schematic illustrating the primary structure of the full-length SARS-CoV-2 spike (S) protein with color-coded domains: N-terminal domain (NTD), receptor binding domain (RBD), fusion peptide (FP), heptad repeat 1 and 2 (HR1, HR2) domain, transmembrane domain (TM), and cytoplasmic tail (CT). The S protein comprises two subunits, S1 and S2. Asparagine residues crucial for N-glycosylation regulation of S binding to the angiotensin-converting enzyme 2 (ACE2) receptor on the host cell are highlighted based on their sequence position. (**B**). Illustration depicting the mechanism of trimeric S glycoprotein binding to human ACE2, initiated by at least one protomer’s RBD switching from a ‘down’ (closed) state (unfavorable for ACE2 binding) to an ‘up’ (open) state (favorable for ACE2 binding). The S protein is represented by cartoons, with individual protomers colored in cyan, red, and gray. Both the ‘down’ closed state (PDB: 7ZH2) and ‘up’ open state (PDB: 7ZH5) of the S protein RBD are shown. Glycans are depicted in blue using Van der Waals representation. N-glycans at positions N165 and N234 are identified as essential structural elements for maintaining the SARS-CoV-2 spike protein in a conformation conducive to ACE2 recognition, facilitating subsequent viral entry into host cells. The glycan at N343 lifts and stabilizes the RBD throughout the opening transition. Visualizations of the S protein were generated using Visual Molecular Dynamics (v1.9.4a55).

**Figure 2 biomolecules-14-00282-f002:**
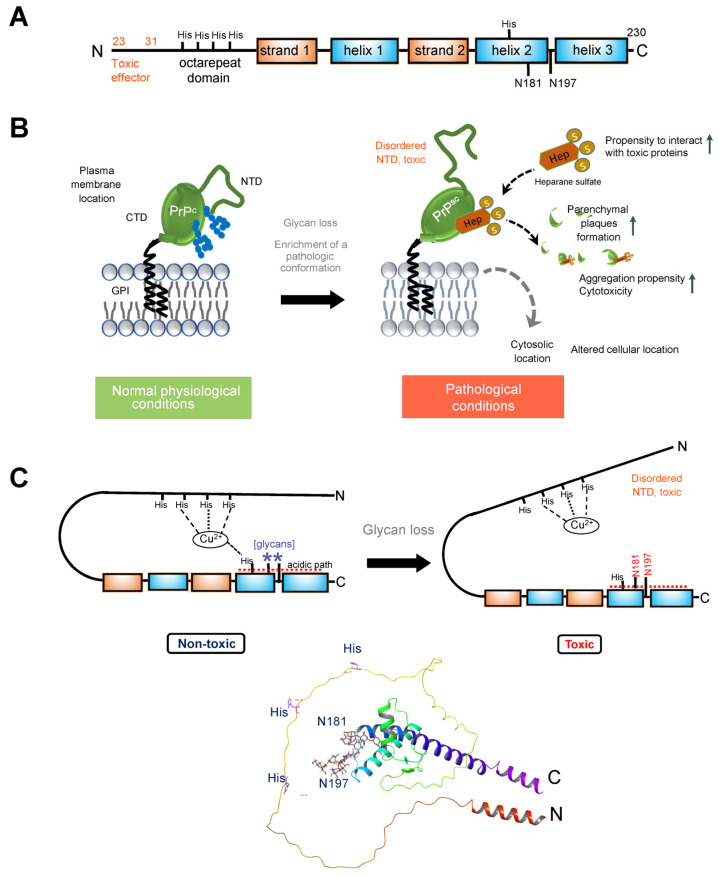
(**A**). Schematic illustrating the primary structure of the full-length prion (PrP^c^) protein. The N-terminal segment (residues 23–125, after removal of the signal peptide) exhibits a high degree of flexibility. Within this segment is an octapeptide repeat domain that binds divalent ions. The C-terminal domain (residues 126–230) folds to a characteristic structure composed of three α-helices, numbered one through three (or A through C), and two anti-parallel β-strands flanking helix 1. PrP^c^ is modified by N-glycans at positions Asn181 and Asn197 in the human protein (corresponding to Asn180 and Asn196 in the mouse protein). Several histidine residues implicated in regulating PrP^c^ stability are also indicated. (**B**). PrP^C^ is attached to the outer leaflet of the plasma membrane through a glycosyl phosphatidylinositol (GPI) moiety. Loss of glycans destabilizes the prion protein structure, enriching it in a conformation that enables pathologic interactions (such as with heparane sulfate). The fewer glycosylation modifications PrP undergoes, the more likely it is to be located in the cytoplasm and the stronger its proteolytic resistance, toxicity, and aggregation ability. (**C**). **Top**, cartoon showing how glycans stabilize the PrP protein. Glycosylation at Asn197 may have an allosteric effect, with impact on participation in stabilizing a stable conformation of the protein. N181 may act in concert with other residues to anchor the disordered NTD to its regulatory CDT. Specifically, Histidine residues, acting through Cu^2+^ coordination, glycans at Asn181, and acidic residues (see acidic patch) may act in concert to anchor the toxic effector N-terminal domain to its regulatory site on the C-terminal domain. **Bottom**, ribbon representation of the PrP^C^ showing the position of the two glycans (sticks). AlphaFold-generated structure using the human prion protein sequence (UNIPROT ID: P04156).

**Figure 3 biomolecules-14-00282-f003:**
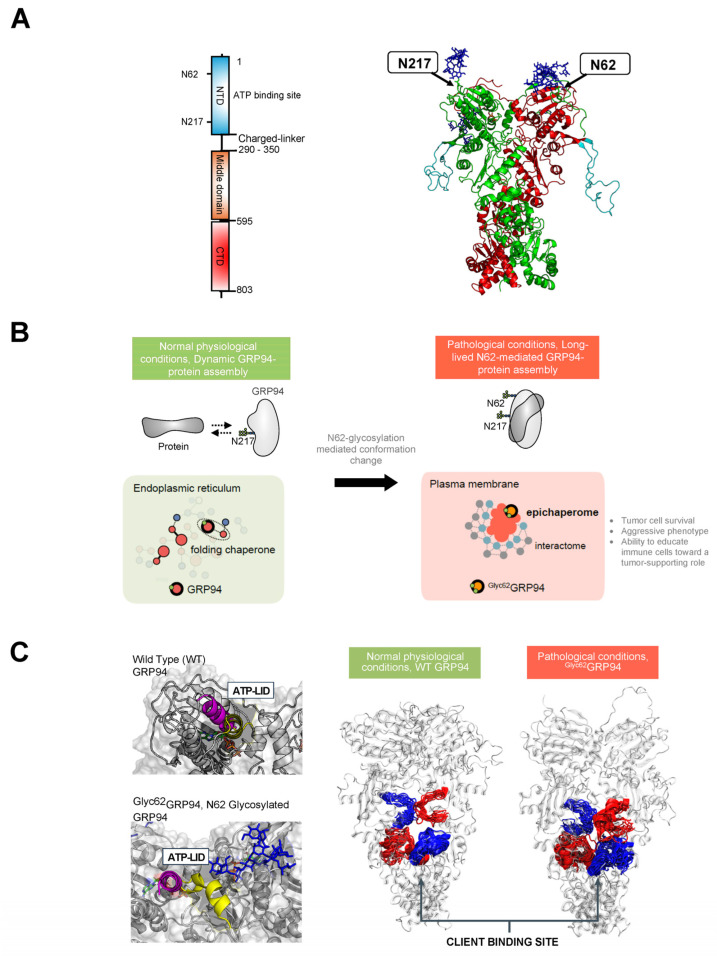
(**A**). Schematic illustrating the primary structure of the full-length GRP94 protein with color-coded domains: N-terminal domain (NTD), middle domain, and C-terminal domain (CTD). The right-side figure depicts the 3D structure of the GRP94 dimer, with individual protomers colored in green and red, respectively. An unstructured loop is colored in light blue. The location of two key regulatory N-glycans, on asparagine residues N217 and N62, is indicated by an arrow. (**B**). GRP94 glycosylation at N62 is a pivotal factor turning GRP94 from a folding chaperone to a pathologic protein. Primarily, N-glycosylation at N62 serves as a structural mediator, inducing the conversion of the GRP94 chaperone into epichaperomes—hetero-oligomeric forms tightly composed of chaperones, co-chaperones, and other factors. In these epichaperomes, GRP94 adopts scaffolding functions not observed in normal cells, where GRP94 primarily participates in protein control and folding. Through this scaffolding function, GRP94 influences the assembly and connectivity of proteins crucial for maintaining a malignant phenotype, enhancing their activity. Ultimately, the N62 glycosylation affects GRP94’s structural ensembles and its chaperone cycle kinetics, leading to a remodeling of the interactome at a much larger scale than one might hypothesize based on a simple local covalent modification. This malfunction in GRP94 amplifies its impact beyond immediate interactors, extending to the remodeling of cellular phenotypes. (**C**). The figure illustrates the structural impact of N62 glycan on GRP94. In the absence of the N62 glycan, GRP94 adopts a conformation favorable for its folding function, as evidenced by the lid of the ATP binding site (depicted in purple) and a more open, flexible protein client binding site (depicted as red or blue ribbons for protomer A or B, respectively). In the N62 glycosylated GRP94, the glycan pulls the ATP-lid into an open conformation (yellow), favoring a more closed and inaccessible protein client binding site that is thus unfavorable for folding. Through its impact on the ATP binding site’s efficiency and by perturbing GRP94’s structural dynamics, N62 glycosylation thus actively shifts GRP94 from a foldase to a protein-assembly platform.

**Figure 4 biomolecules-14-00282-f004:**
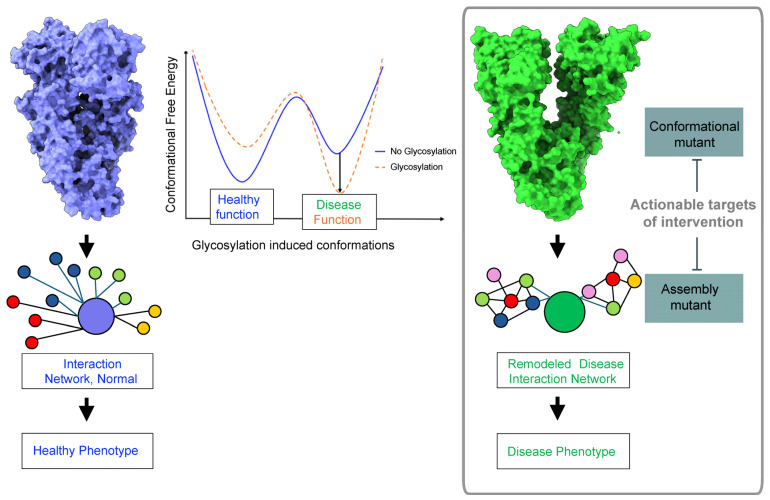
The schematic illustrates the central role of N-glycans in orchestrating protein conformation and assembly dynamics, illustrating their profound impact on disease. The figure emphasizes that glycosylation-induced conformational mutants, characterized by altered three-dimensional structures or conformational dynamics compared to wild-type proteins, serve as unique targets for therapeutic intervention. Importantly, the figure highlights how glycan-mediated remodeling extends beyond individual protein structures, influencing the interactome of the protein and reshaping functional pathways at the systems level. This intricate interplay at the molecular and systems levels ultimately contributes to the reshaping of cellular phenotypes. The schematic not only highlights how N-glycosylation provides actionable targets for precision therapies but also underscores the transformative potential of modulating glycan-regulated protein structures and functions in a systemic context.

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
