# Peer review of "N-Glycosylation as a Modulator of Protein Conformation and Assembly in Disease"

_biomolecules, 2024, doi:10.3390/biom14030282_

Round 1

Reviewer 1 Report

Comments and Suggestions for Authors

The presented review is a very valuable article, the authors extensively summarize issues related to the interplay of glycosylation and protein conformation, with particular emphasis on the profound impact of N-glycans on the selection of distinct protein conformations under normal and pathological conditions in various diseases. The manuscript is carefully prepared, there is no need to introduce significant changes or corrections. It contains all the necessary elements, the presented issues are fully developed and thoroughly described, detailed conclusions and future directions are also included. Additionally, the manuscript contains an extensive literature list, pertinent and adequate references to related works. Minor corrections are mentioned below.

I would like to suggest a few comments:

1.       The two types of glycans should be clarified in this sentences “Consequently, these two glycans were identified as contributors to stabilizing both ‘down’ and ‘up’ states, establishing a local energy minimum for each.” line 349-350

2.       Explanations for abbreviations should be introduced: Neu5Gc - line 115, OST - line 624

3.       The text in Figures 1B, 2B is in a light shade and is therefore difficult to see when printed.

4.       I would add a citation to this statement: “It is estimated that more than 50% of mammalian proteins are glycosylated, emphasizing the prevalence and significance of this modification.” line 145

5.       This sentence seems too long, I would suggest dividing it into two parts “Among the well-studied and understood examples are antibodies, where changes in glycan composition at specific sites are known to influence the antigen-binding affinity of monoclonal antibodies, with fucosylation reducing this interaction [61-65]; the cancer cell glycocalyx—a layer of multifunctional glycans that covers the cell surface—where abundant glycosylation, including sialylation, fucosylation, O-glycan truncation, and N- and O-linked glycan branching has an impact on cell adhesion, and promotion of cancer migration and invasion [18,66,67].”

6.       I would suggest clarifying what and how the two types of glycans play a crucial role in maintaining an intramolecular interaction with the N-terminal domain? “NMR and electron paramagnetic resonance spectroscopy studies suggest that the two N-glycans play a crucial role in maintaining an intramolecular interaction with the N-terminal domain [111,116,117]”. line 531

7.       In the sentence “Additionally, PrP glycans promote the N-C interaction, synergizing with the effect of His-Cu coordination [111,118]. “ should be clarified His-Cu2+ coordination or His-Cu (II) coordination;  line 537

Author Response

Response: We are pleased to learn that Reviewer 1 found the review to be comprehensive and well-prepared. Thank you for your constructive feedback, and the opportunity to incorporate your suggestions into the final version of our manuscript.

I would like to suggest a few comments:

  1. The two types of glycans should be clarified in this sentences “Consequently, these two glycans were identified as contributors to stabilizing both ‘down’ and ‘up’ states, establishing a local energy minimum for each.” line 349-350

Response: done as suggested

  1. Explanations for abbreviations should be introduced: Neu5Gc - line 115, OST - line 624

Response: done as suggested

  1. The text in Figures 1B, 2B is in a light shade and is therefore difficult to see when printed.

Response: color of the font was changed as suggested and figures were provided as a separate file, combining the four figures – CombinedFigures -F-revised.pdf

  1. I would add a citation to this statement: “It is estimated that more than 50% of mammalian proteins are glycosylated, emphasizing the prevalence and significance of this modification.” line 145

Response: a reference was added as suggested

  1. This sentence seems too long, I would suggest dividing it into two parts “Among the well-studied and understood examples are antibodies, where changes in glycan composition at specific sites are known to influence the antigen-binding affinity of monoclonal antibodies, with fucosylation reducing this interaction [61-65]; the cancer cell glycocalyx—a layer of multifunctional glycans that covers the cell surface—where abundant glycosylation, including sialylation, fucosylation, O-glycan truncation, and N- and O-linked glycan branching has an impact on cell adhesion, and promotion of cancer migration and invasion [18,66,67].”

Response: edited as suggested

  1. I would suggest clarifying what and how the two types of glycans play a crucial role in maintaining an intramolecular interaction with the N-terminal domain? “NMR and electron paramagnetic resonance spectroscopy studies suggest that the two N-glycans play a crucial role in maintaining an intramolecular interaction with the N-terminal domain [111,116,117]”. line 531

Response: the text was edited to provide clarity, as suggested

  1. In the sentence “Additionally, PrP glycans promote the N-C interaction, synergizing with the effect of His-Cu coordination [111,118]. “ should be clarified His-Cu2+coordination or His-Cu (II) coordination;  line 537

Response: the text was edited for clarity

Reviewer 2 Report

Comments and Suggestions for Authors

This is a timely review that gives a fresh perspective on the complexity of the interactions of complex glycans with receptor proteins and its biological consequences. The authors demonstrate the intricate interplay between glycosylation and physiological impact by biologically relevant examples. This review will help to identify starting points for future work on unravelling the complex nature of glycan-protein interactions in biological systems. The review is well structured and deserves publication.

Author Response

This is a timely review that gives a fresh perspective on the complexity of the interactions of complex glycans with receptor proteins and its biological consequences. The authors demonstrate the intricate interplay between glycosylation and physiological impact by biologically relevant examples. This review will help to identify starting points for future work on unravelling the complex nature of glycan-protein interactions in biological systems. The review is well structured and deserves publication.

Response: We sincerely appreciate Reviewer 2’s thoughtful review and positive feedback on our manuscript. The insights and encouraging comments are invaluable to us.